# The European Stag Beetle (*Lucanus cervus*) Monitoring Network: International Citizen Science Cooperation Reveals Regional Differences in Phenology and Temperature Response

**DOI:** 10.3390/insects12090813

**Published:** 2021-09-10

**Authors:** Arno Thomaes, Sylvie Barbalat, Marco Bardiani, Laura Bower, Alessandro Campanaro, Natalia Fanega Sleziak, João Gonçalo Soutinho, Sanne Govaert, Deborah Harvey, Colin Hawes, Marcin Kadej, Marcos Méndez, Bruno Meriguet, Markus Rink, Sarah Rossi De Gasperis, Sanne Ruyts, Lucija Šerić Jelaska, John Smit, Adrian Smolis, Eduard Snegin, Arianna Tagliani, Al Vrezec

**Affiliations:** 1Research Institute for Nature and Forest (INBO), 1000 Brussels, Belgium; 2Independent Researcher, 2000 Neuchâtel, Switzerland; barbalat.richard@bluewin.ch; 3Raparto Carabinieri Biodiversità di Verona, Centro Nazionale Carabinieri Biodiversità ‘Bosco Fontana’, 46045 Marmirolo, Italy; bardianimarco@gmail.com; 4People’s Trust for Endangered Species, London SW8 4BG, UK; laura.bower@ptes.org; 5CREA, Centro di Ricerca Difesa e Certificazione, 50023 Firenze, Italy; alessandro.campanaro@crea.gov.it; 6Department of Biosystems (BIOSYST), Catholic University of Leuven, 3000 Leuven, Belgium; natalia.fanegasleziak@kuleuven.be; 7Associação Bioliving, 3850-635 Frossos, Portugal; soutinhojg@gmail.com; 8CIBIO/InBIO—Centro de Investigação em Biodiversidade e Recursos Genéticos, Universidade do Porto, 4485-661 Vairão, Portugal; 9Faculdade de Ciências, Departamento de Biologia, Universidade do Porto, 4169-007 Porto, Portugal; 10Faculty of Bioscience Engineering, Department of Environment, Forest & Nature Lab, Ghent University, 9000 Ghent, Belgium; Sanne.Govaert@UGent.be; 11Department of Biological Sciences, Royal Holloway University of London, Egham TW20 0EX, UK; d.harvey@rhul.ac.uk; 12Independent Researcher, Bentley IP9 2BS, UK; hawescolin@gmail.com; 13Department of Invertebrate Biology, Evolution and Conservation, University of Wrocław, 51-148 Wrocław, Poland; marcin.kadej@uwr.edu.pl (M.K.); adrian.smolis@uwr.edu.pl (A.S.); 14Area of Biodiversity and Conservation, Rey Juan Carlos University, 28933 Móstoles, Spain; marcos.mendez@urjc.es; 15Office Pour Les Insectes et Leur Environnement, 78280 Guyancourt, France; bruno.Meriguet@insectes.org; 16Hirschkäferfreunde-Nature Two e.V., 56859 Alf, Germany; hirschkaefer-rink@t-online.de; 17Department of Biology-Natural History Museum ‘La Specola’, University of Florence, 50125 Florence, Italy; sarahrodega@gmail.com; 18Natuurpunt, 2800 Mechelen, Belgium; sanne.ruyts@natuurpunt.be; 19Faculty of Science, Department of Biology, University of Zagreb, 10000 Zagreb, Croatia; slucija@biol.pmf.hr; 20European Invertebrate Survey—The Netherlands/Naturalis Biodiversity Center, 2300 Leiden, The Netherlands; john.smit@naturalis.nl; 21Biology and Chemistry Department, Belgorod State University, 308015 Belgorod, Russia; snegin@bsu.edu.ru; 22Department of Earth and Environmental Sciences, University of Pavia, 27100 Pavia, Italy; arianna.tagliani01@universitadipavia.it; 23Department of Organisms and Ecosystems Research, National Institute of Biology, 1000 Ljubljana, Slovenia; al.vrezec@nib.si

**Keywords:** transect walk, monitoring protocol, drop-out, habitats directive, volunteers, threatened species

## Abstract

**Simple Summary:**

International cooperation is needed to prevent the loss of threatened species. To evaluate the situation, standardised monitoring is an important tool. Involving the general public (citizen science) can play a crucial role in realising such international monitoring. Here we report on the start-up and initial findings of the European Stag Beetle Monitoring Network (ESBMN), an international network of stag beetle (*Lucanus cervus*) monitoring schemes using the same protocol. This network aims to regularly assess local and international changes in the population of the stag beetle. Therefore, an internationally standardised protocol was agreed and a website was created where volunteers can create a transect and submit the data of their transect walks. Currently, the number of transects and transect walks submitted is increasing annually and will soon allow trend analysis. Our initial experience with the ESBMN shows that volunteers need more guidance and encouragement to avoid them dropping out of the project. In conclusion, we believe this system of international cooperation can be used for other charismatic insects in order to evaluate their threatened status and plan conservation actions.

**Abstract:**

To address the decline in biodiversity, international cooperation in monitoring of threatened species is needed. Citizen science can play a crucial role in achieving this challenging goal, but most citizen science projects have been established at national or regional scales. Here we report on the establishment and initial findings of the European Stag Beetle Monitoring Network (ESBMN), an international network of stag beetle (*Lucanus cervus*) monitoring schemes using the same protocol. The network, started in 2016, currently includes 14 countries (see results) but with a strong variation in output regarding the number of transects (148 successful transects in total) and transect walks (1735). We found differences across European regions in the number of stag beetles recorded, related to phenology and temperature, but not for time of transect start. Furthermore, the initial experiences of the ESBMN regarding international cooperation, citizen science approach, and drop-out of volunteers is discussed. An international standardised protocol that allows some local variation is essential for international collaboration and data management, and analysis is best performed at the international level, whereas recruiting, training, and maintaining volunteers is best organised locally. In conclusion, we appeal for more joint international citizen science-based monitoring initiatives assisting international red-listing and conservation actions.

## 1. Introduction

Insects have undergone a massive decrease in population abundance and biodiversity in recent decades [1,2,3]. Long-term monitoring schemes are essential if we are to understand the causes and current state of biodiversity loss, identify the main drivers, and plan mitigation strategies and thorough conservation actions [1,4]. To accurately evaluate the biodiversity crisis, monitoring has become an increasingly important tool [5], e.g., international nature conservation acts such as the European Habitats Directive include regulations for monitoring programmes.

To operate these monitoring programmes, and for the assessment of International Union for Conservation of Nature (IUCN) red list status, a citizen science procedure has often been used [6,7,8,9]. Most citizen science projects are implemented at local or national scales [10], but see [11]. This may be logical due to the level at which monitoring data are required and at which potential volunteers are associated with non-governmental organisations (NGO) and fora. Firstly, however, conservation policies and funding opportunities are shifting from national to international level (e.g., [12,13,14,15]). Secondly, international comparison of national datasets is often not possible due to differences in monitoring protocols, or is limited to comparing metadata [16,17]. Thirdly, trend analysis can be demanding in terms of the number of monitoring sites and volunteers involved [18,19,20]. These may often not be met for rare species or small regions or countries (e.g., [20]), and data at large spatial scale is needed [21]. 

Red list initiatives for insects have also recently been scaled up from national to international level (e.g., [22]), but monitoring data on insects to analyse population trends at the international level are still very scarce [7]. Citizen science may potentially help to gather this information, but international citizen science projects, such as the European Butterfly Monitoring Scheme [23], are still limited. 

A similar monitoring network was set up for the European stag beetle (*Lucanus cervus*, Coleoptera: Lucanidae). This species is an ideal target species for citizen science due to its large size, charismatic appearance, pan-European distribution, and easy identification [8,24]. Consequently, there is a long history in using citizen science to compile data regarding this species [25]. In many areas it is part of urban wildlife [26] making a citizen science approach more feasible [10,27]. Moreover, the species is protected by the European Habitats Directive (Annex II) and evaluated as a species of Least Concern [22] due to its patchy distribution. Furthermore, this species is often considered to be an umbrella species for dead wood conservation, making it interesting for citizen science projects and educational programmes [26,28]. 

In 2012, a standardised monitoring protocol was developed, encompassing trials at 29 transects in eight countries [28]. Furthermore, a statistical power analysis was performed based on 1650 transect walks from three transects (UK, Germany, and Switzerland) revealing the need for 240 transect walks per year, from 40 to 100 transects, in order to reveal small population trends [20]. This number of transects (and volunteers) may be unfeasible for many regions and small countries due to limited funding, availability of volunteers, or even the number of stag beetle populations present. Consequently, international cooperation is needed to gather sufficient data for robust and sound statistical trend analysis. International cooperation is also crucial to achieve integrated coordination of methodology, education, aims, and communication [14,15]. With these ideas and pilot studies in mind, the European Stag Beetle Monitoring Network (ESBMN) was set up in 2016.

The goal of this paper is to describe the ESBMN, where we tried to maximise the advantages of international cooperation with national or regional representatives, to organise a citizen science project with international data collection for evaluation of the population decline in the stag beetle. This is demonstrated by initial results regarding data collection and international cooperation. As data mainly come from three separate regions in Europe, we analyse the effect of these regions on the species phenology, temperature response, and transect starting time. These three parameters are known to strongly influence the number of observations [20,28]. Finally, we discuss lessons learned from establishing the ESBMN regarding the aspects that are best organised at local versus international level to guide similar future cooperation initiatives. 

## 2. Materials and Methods

### 2.1. ESBMN and Monitoring Protocol

The ESBMN was established in 2016 as an international network of organisations involved in national or regional stag beetle monitoring schemes (further referred to as country representatives). It was built on the basis of previous informal scientific collaboration among some of the participants. The network currently includes seven universities, four governmental research institutes or administration bodies, and six NGOs involved in the research, conservation, and monitoring of this species in 14 countries. The ESBMN mission is to realise a population density change monitoring of the European stag beetle with the aim to regularly assess local and international population changes. Our vision is to establish a monitoring network with a uniform methodology, allowing a citizen science approach, and to assess the species status at the full scale of its range. The strategic plan and data policy are available on the website [29]. It is an open network allowing the addition of new members that are working on a local monitoring network. The ESBMN provides a standardised monitoring protocol and a website (www.stagbeetlemonitoring.org) (accessed on 1 August 2021) that allows entering, storing and managing data, and exchange of best practices in setting up a network of transects between members and cooperating in scientific analyses and publications. Specifically for the volunteers and general public (479 subscribers), ESBMN provides a yearly newsletter communicating the conclusions of the monitoring, in addition to general knowledge on the species, and raising awareness about the threats faced by this species.

The standardised protocol (Appendix A) is translated into and published in nine languages on the ESBMN website. In summary, volunteers choose a 500 m transect where stag beetles are known to occur which is walked each time in the same direction. Volunteers are asked to walk the route at least six times per year during the months of June and July on warm (>12 °C) evenings with little or no rain and no strong wind. Transect walks start about 15 min before sunset at a gentle pace taking 30 min to complete the 500 m. Observations within a virtual box of approximately 10 m in front, and 5 m to each side, of the surveyor (Figure 1) are registered (cf. [30]). For each observation, the time, sex, and activity of the beetle is noted. This protocol is mainly based on an earlier version [28]. A limited number of changes were discussed among the members to simplify the protocol.

The country representatives oversee outreach, training, and communication to volunteers at regional or national scales. Volunteers directly enter the data onto the ESBMN website. The manner in which organisations promote this network can differ (see results). Furthermore, some countries have a slightly different monitoring protocol. In Italy, the transects are walked partly by professionals and partly by citizen science volunteers, and most transects are walked alternately back and forth and dead specimens are not noted [31]. In Slovenia and Croatia, all transects are walked by professionals and the length differs based on the location (ranging from 100 to 1000 m). These transect walks start around sunset and the length of transect walks depends on the length of the transect, varying from 20 to 60 min [32,33]. Transect data from Italy, Slovenia, and Croatia are not entered directly on the website but added on a yearly basis to the ESBMN database. Finally, the only Swiss transect and one UK transect are point transects, counting stag beetles during 30 min from a single viewpoint. This point monitoring is currently implemented as an additional monitoring scheme. 

All transects were verified and classified as standardised line transects (400–600 m), or a point, shorter, or longer transect. All transect walks were also checked and labelled as “according to protocol (started between 1:15 before to 0:45 after sunset)”, early (2:15 to 1:15 before sunset), late (0:45 to 9:45 after sunset), daytime (14:15 to 2:15 before sunset), or using the Italian protocol [31]. A few submissions were also labelled as “not according to protocol”. 

### 2.2. Statistics

Analysis was based on all transects but only transect walks “according to protocol” were included (see Appendix A for the data). Furthermore, 234 transect walks were omitted with missing data. Based on the geographical spreading, transects were grouped into Southwest (Portugal, Spain, and France; 503 transect walks after omitting records with missing data), Southeast (Switzerland, Italy, Slovenia, and Croatia; 300 transect walks), and North (United Kingdom, Belgium, The Netherlands, Germany, and Poland; 592 transect walks). Between transects from different regions, there is at least 340 km (Figure 2). The Polish transects were at least 540 km from any other transect and have a much more continental climate; however, there were too few transect walks to analyse them separately. In addition to regional differences, the Northern transects are all in lowland areas, whereas the Southeast and Southwest transects include both lowland and upland transects. Other variables used for the statistical analyses were seasonal effects, temperature, and start time of the transect walk. To include the seasonal effect, the date of each transect walk was converted to day-of-year (doy). To avoid including large numbers in the model, doy was divided by 30, which can be interpreted as month-of-year (moy) (cf. [20]). Temperature was measured by the volunteer at the start of the transect walk and expressed as degrees Celsius. Finally, the start time was expressed in minutes before sunset. 

To evaluate the regional effect on the elements known to influence the number of observations [20,28,31], a statistical model was built. Due to the non-linear nature of the responses (e.g., phenology), a generalised additive model (gam) was built with a separate spline smoother for moy, temperature, and starting time. The three regions were used as the interaction term for each smoother. The model can be written as:g(E(No. observations)) = s(moy: region) + s(temperature: region) + s(start time: region)(1)
with g representing the log-link function and s representing the spline smoothers. The position of the transect (defined by latitude and longitude) was included as a spatial Gaussian random effect. A negative binomial distribution was used for the number of observations. We fit the model in R 4.0.2 [34] using the mgcv package [35].

## 3. Results

The current network includes 14 countries, which entail 38% of the countries (excluding city states) within the range and 71% of the range surface (approx. 5,700,000 km^2^, Figure 2). 

To date, 195 transects were set up in a total of 13 countries (Table 1, Figure 2). A strong drop-out effect can be detected when comparing the number of volunteers registered (656), the number of transects created (195), the number of transects with at least one transect walk (148), and the number of transects with transect walks in multiple years (54). Because the network was established in 2016, data is mainly limited to the period 2016–2020. In most countries, the initiation of this monitoring is relatively recent, resulting in a limited number of years with data for each transect. Monitoring was only started prior to the establishment of the network in Belgium and Slovenia, resulting in earlier data in these countries and, consequently, higher mean years of data per transect. The 195 transects include 181 standard line transects, two point transects, nine shorter transects, and three longer transects. From the 1735 transect walks, 1538 used the standardised protocol, 93 the Italian protocol, 71 were walked earlier, 26 during daytime, and seven late. The annual number of transect walks increased from 93 in 2016 to 345 in 2019 and 656 in 2020. 

The largest number of transects established are found in the UK (50) and Portugal (46), but these countries also have the largest drop-out effect for transects with multiple years of data (91% and 90%) and, consequently, only four and one transects, respectively, have been monitored in multiple years. The highest number of transects with multiple years are found in The Netherlands (12), Slovenia (10), and Spain (10). The lowest drop-out rates are found in Slovenia (0%), Switzerland (0%, only one transect), Germany (25%, only four transects), The Netherlands (40%), and Croatia (40%). To date, no transects have been established in Ukraine and the Russian Federation.

The number of observations was significantly influenced by most model parameters, with the exception of the starting time and the temperature in the Southwest region (Table 2). The phenology appeared to be later in the Southeast region (optimal period in early July) compared to the North and Southwest region (optimal period in mid-June to late June, respectively, Figure 3). The model estimated a slightly longer season in the North and Southwest, with 75% and 79%, respectively, of the observations in June and July (compared to May–August) vs. 89% in the Southeast. 

A positive effect of temperature on the number of observations was found in all three regions, but was not significant in the Southwest (Figure 4). In the Southeast, the temperature response appears initially slightly stronger, whereas for high temperatures (above 25 °C) a decrease in activity is seen. In the North, the response remains high after 25 °C but the confidence intervals are widened. 

For starting time, the effect was not significant for all of the three regions. Figure 5 indicates that more observations are noted on transect walks before sunset in the Southwest, whereas the opposite is true for the North. In the Southeast, there may be an optimum about 20 min before sunset. 

In addition to the effect of the three regions, an important effect of longitude and latitude remained in the model as a random effect. This random effect takes up spatial variability between transects that is not explained by other model parameters, such as local habitat suitability or climate.

## 4. Discussion

The ESBMN was established to promote a standardised monitoring enabling a citizen science approach. To date, 14 countries have been included, resulting in 148 successful transects and 54 transects that have been monitored during different years. Data from 1735 transect walks have been collected. A preliminary analysis indicated differences across European regions in the number of stag beetles recorded, related to phenology and temperature, whereas the time of transect start did not show any significant effects. Below, we first discuss the results of the analysis. Second, we talk about the major implications of our experiences for the organisation and consolidation of this and similar international networks. Finally, we discuss the challenges and benefits of a citizen science approach.

### 4.1. Results of the Analysis

We found that phenology in the Southeast transects was later than that in the Southwest and North. Differences in phenology were previously documented at the local scale related to altitudinal differences [36,37,38], but across Europe differences have not yet been found [20,28]. Schmucki et al. [39] found a later phenology for butterflies in cold and mesic climates, which coincides with many Southeast transects. At present, no firm conclusions can be drawn for these differences, which might be related to regional differences, in addition to differences in altitude or climate. To fully explore these effects, more data may be needed, and actual altitude and climatic variables need to be included in the model. For temperature, regional differences may be explained by climate where colder regions have a stronger response to temperature as warm evenings are less frequent and activity is consequently concentrated during these evenings. Within the variability studied, 1 h 15 min before to 45 min after sunset, starting time had no significant effect on the number of observations. This means that starting time does not need to be strictly fixed in the protocol. Moreover, the stag beetle activity may be strongly related to the amount of light (thus civil twilight) than the time of sunset. The time between civil twilight and sunset increases Northwards, which may explain the pattern observed.

### 4.2. Implications of Our Experiences

The goal of the ESBMN is to assess the trend of the species. Because the ESBMN was established in 2016, it is too early to analyse the data for any trends. Nevertheless, the annual number of transect walks gathered has increased strongly. Because the initiatives in most countries were only recently established, or have still to be fully elaborated, similarly strong growth can be expected in the future. At an international level, we have already achieved the goal of 240 transect walks from 40 transects per year, as suggested for long-term trend analysis [20]. Two of the three regions analysed are at or close to this level (Southwest: 354 transect walks from 51 transects and North: 188 transect walks from 28 transects in 2020). At the national level, we are still far from this level of effort, with the exception of Portugal (249 transect walks from 36 transects in 2020). Because there is no strict commitment, strong differences between countries’ engagement may also remain in the future. However, if the general effort can be maintained or slightly improved during the next few years, it is feasible that sufficient years of data will be collected, which would thus enable trend analysis. Current modelling techniques allow us to cope with this kind of unbalanced data set. Consequently, a difference in engagement or start-up year between countries should not be a concern for future analysis. Furthermore, innovative techniques allow trend or population estimates to be separately made for each transect (or group of transects) [39,40,41]. 

From the experience of the first years of activity of the ESBMN, we learned that international cooperation has clear benefits. Elemental was the development of an international standardised monitoring protocol to ensure data comparison, in addition to saving development time and management costs, and avoiding repeated flaws [15,42,43]. Because the species phenology differs across its range, some elements of the protocol need local variation to optimise the number of observations. Moreover, the organisation of the network needs some flexibility to fit local requirements and opportunities. In our example, the organisation ranges from social media campaigns to professionally monitored transects. The first have been used in Portugal and UK to find volunteers that install their own transect. This approach has yielded high responses, but is also likely to be related to a large drop-out rate in these two countries. In Flanders (Northern Belgium) and The Netherlands, where a limited number of populations remain, transects have been set up by the country representatives, and volunteers are actively sought for these transects. In Spain, the country representative mainly takes care of disseminating the protocol whereas local NGOs, county administrations, and individuals have been involved in the actual monitoring. Finally, the transects monitored in Slovenia and Croatia are followed up by employees of the research institute and national parks, respectively, where the drop-out effect is consequently low. In other countries, the campaign is limited or only recently initiated. A similar variety in organisation exists in the European Butterfly Monitoring Scheme [23]. Data management is another element that is best organised at the international level in order to maximise opportunities for data exchange and comparison [10]. Because detailed data analysis can be demanding in terms of the number of records needed [20,44], the analysis is also best performed at the international level whenever possible. Because potential errors and biases are linked to citizen science data, the interpretation and relevance of a certain national trend is best evaluated within a larger international framework [10].

In contrast, finding, interacting, and maintaining the volunteers is a task that mainly needs to be organised at national or regional levels, because most volunteers will feel most associated with this level. We send out a yearly newsletter to all volunteers, but this is only in English and may be less relevant for some of the volunteers.

### 4.3. Challenges and Benefits of Our Citizen Science Approach

The main implementation of the ESBMN is based on citizen science monitoring schemes. However, some national or regional schemes completely or partly depend on professionals. In practice, the difference between professionals and volunteers may be significantly smaller than often stated. For example, several citizen scientists within our network have an academic career or work in nature conservation. By comparison, forest guards may be seen as professionals but sometimes have no experience with the species.

The main difference may be that it requires a large investment to find and maintain volunteers [42], whereas the organisation of professionals appears, in general, to be easier and the long-term sustainability of a fixed executer for each transect is more likely to be ensured. The large disadvantage is that the number of professional employees is limited, even when broadened to forest guards and local conservationists. This is especially important for this species, which is only active during a short period in the evening, allowing only one transect walk per person per day. In Slovenia, 10 transects are conducted by a team of four researchers, whereas in Croatia, to date, five transects have been established in four national parks by local employees. In other countries the use of professionals is sporadic. With professionals alone, the effort may not be sufficient to separately evaluate trends from these data. Furthermore, employees may not have the time or interest to take up this additional task and work during the evening (outside standard office hours). It has been repeatedly found that there are no differences in data quality between both groups if the monitoring is straightforward and the species can be easily identified (e.g., [45,46,47]). Consequently, we believe that the data quality in this network will be comparable. Finally, professional monitoring can be combined with the monitoring of other species. which makes the task more complex but also yields a higher return in results.

The type of transects that professionals and volunteers cover might also differ. Volunteers will often walk transects near their home and, consequently, are well placed to monitor urban or residential populations [10]. In contrast, transects walked by professionals are more often characterised by large populations in natural and protected areas. Consequently, whether the best strategy is to rely on either volunteers or professionals, or a combination of both transect types, might differ depending on the local habitat. This is especially relevant to ensure a representative trend because it may differ between urban and protected areas. A citizen science approach, furthermore, helps with public awareness because some volunteers can gradually also take up conservation or dissemination tasks [42]. By contrast, local managers can use the data directly for local conservation purposes. 

The dropout effect has often been considered as an important drawback of citizen science, especially for long-term monitoring (e.g., [44]). During our implementation we have also seen large dropouts in several countries. The main reasons for dropout effects found include: (1) insufficient information about the scheme; (2) limited feedback and engagement; (3) limited number/diversity of observations; (4) limited appreciation and communication building; and (5) limited training and guidance [10,42,43,44,48]. Many of these elements have also been found to limit data quality (see further) [43]. Up to a certain level, all of these reasons may explain a portion of our dropout effects, but the effect might differ between countries and campaigns. Furthermore, drop-outs may be specifically related to the period (late June and early July), which partly coincides with the main holiday period when many volunteers are unavailable. Country representatives have begun to take up a much more prominent role in recent years in order to ensure data quality, communicate with volunteers, organise venues and training, give rewards, create volunteer communities, and show the results of the campaign. 

Another important concern for the future is the non-random design in most countries in our network, which may result in an overrepresentation of urban transects or transects with high population densities (cf. [10,23,43]). The transect network is only designed to include all populations in an equal share in Slovenia, The Netherlands, and Flanders. Moreover, the dropouts are likely to be non-random and can further increase the degree of imbalance; for example, when transects with limited observations, which may be most prone to further decline or extinction, have higher dropouts [44]. Possible solutions are (1) looking for new volunteers for discontinued transects, or (2) including professionals for transects with a low number of observations. A post-stratification of transects based on landscape type and protection status can also help to reduce this bias [23]. To improve the influx of new volunteers, it has been highly rewarding to contact volunteers that frequently report single observations of this species because they likely live near a population, are familiar with the species, and are already undertaking to frequently visit the same population (cf. [10]). 

Finally, the data quality is an aspect of citizen science that needs consideration. Reviews of citizen science projects have stressed the importance of good designs, limited complexity regarding protocol and hypothesis, trained volunteers, and professional oversight [10,43]. Regarding the recognition of the species, practical solutions may include a volunteer-indicated certainty level [45], requiring the additional feedback of first year volunteers or excluding the first year data of volunteers. Regarding the protocol, using transect walks to monitor the stag beetle has been repeatedly tested and evaluated both by professionals and volunteers in different regions and habitats, and at different spatial and temporal scales [20,28,31,32,49,50,51,52,53,54]. Thus, we are confident that it is simple but robust enough for the data required. 

Using a similar transect set-up as already used for butterflies facilitates the re-use of methodological and statistical advances. For example, new techniques have been described to evaluate trends of single transects within a larger data frame [39,40], which can now be directly applied to our network as soon as sufficient data are available from several years. 

### 4.4. Perspectives and Conclusions

Our perspectives are that the number of transects and transect walks will steadily grow in the coming years, and that additional countries may join the ESBMN. We will further analyse and compare the organisation of the different schemes in different countries, and learn and improve the schemes accordingly. We hope to further professionalise the guidance of the volunteer community. For the analysis of trends, we will also need to improve the knowledge on potential biases created by the citizen science approach or due to organisational differences between schemes. Due to the growth of the data, the possibility to study and include other covariates will grow and result in better insights in the structure of the data. Secondly, the increased amount of data will improve the coverage of the species range. In the near future, it will be important to align our effort with the potential rise of other monitoring schemes for saproxylic beetles. We also need to look for closer collaboration with European bodies that can help with the continuation of the ESBMN. Because nature conservation monitoring is becoming increasingly dependent on citizen science approaches [42,48,55], integrative approaches (e.g., the European Solidarity Corps and European Citizen Science Association) are strongly welcomed. 

In conclusion, we want to appeal for more joint international citizen science-based monitoring initiatives. In Europe, in particular, national populations often represent a relatively small proportion of the entire population. International standardisation of the monitoring protocol and cooperation can significantly help to gather comparable data at the range level of the species. This will assist international red listing of these species and monitoring of national and international conservation actions. 

## Figures and Tables

**Figure 1 insects-12-00813-f001:**
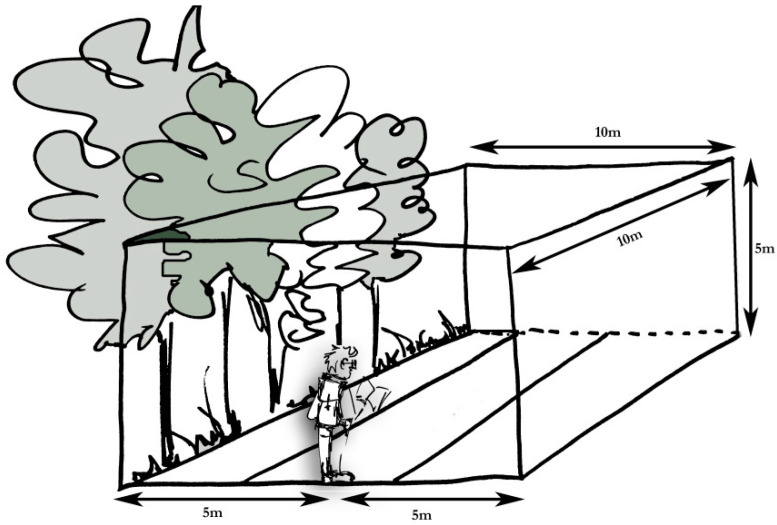
Representation of the virtual box around the surveyor; stag beetles observed within this virtual box are noted. Drawing made by Studiolae.

**Figure 2 insects-12-00813-f002:**
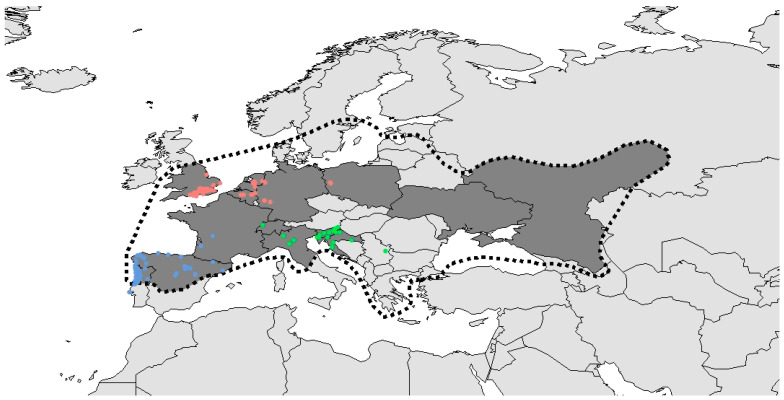
Range (dashed line) of *Lucanus cervus* and monitoring transects (dots). Countries in dark grey are represented in the European stag beetle monitoring network. Different colours of transects represent different regions (see legend Figure 3).

**Figure 3 insects-12-00813-f003:**
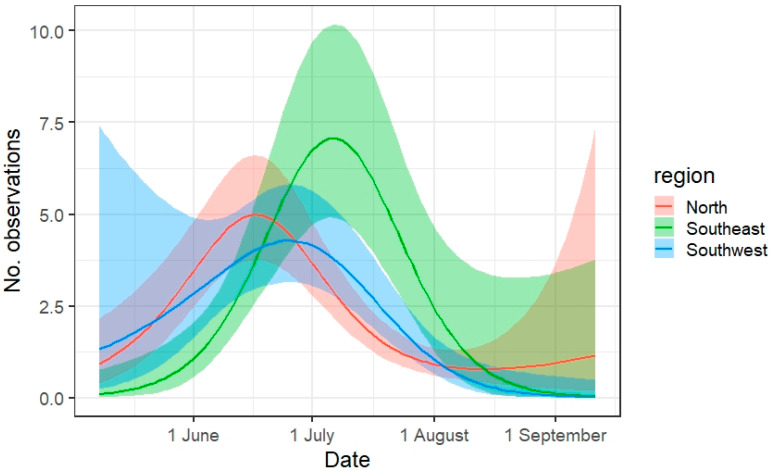
Modelled phenology of the number of observations along a transect walk given for three regions. Model outcome is plotted as a line with the confidential interval as a ribbon, after taking into account the effect of the other variables in the model.

**Figure 4 insects-12-00813-f004:**
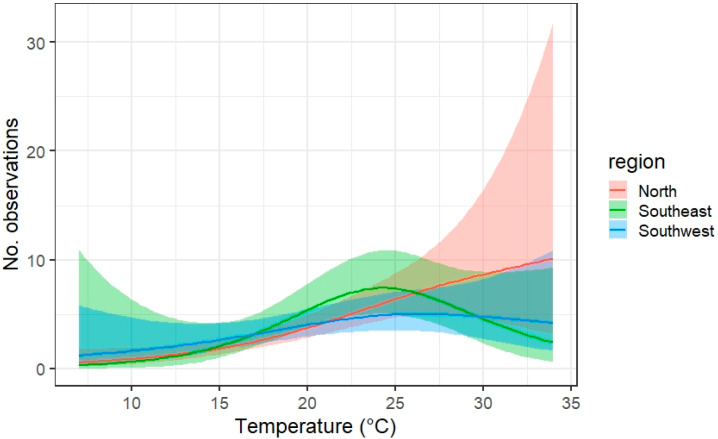
Temperature effect on the number of observations along a transect walk given for three regions. Model outcome is plotted as a line with the confidential interval as a ribbon, after taking into account the effect of the other variables in the model.

**Figure 5 insects-12-00813-f005:**
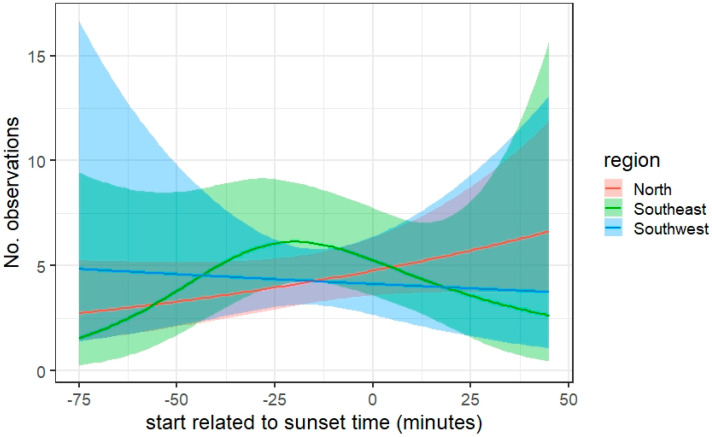
Effect of starting time on the number of observations along a transect walk given for three regions. Model outcome is plotted as a line with the confidential interval as a ribbon, after taking into account the effect of the other variables in the model.

**Table 1 insects-12-00813-t001:** Countries with the number of transects and transect walks ordered by number of transect walks. Percentage comparing numbers to total number of transects set up. For transects with multiple years, percentage is calculated by excluding the transects set up in 2020 because they do not yet have data from multiple years.

Country	No. Transects			Mean Years of Data/Transect	No. Transect Walks
	Set Up (100%)	with at Least One Transect Walk	with Data from Multiple Years		
Slovenia	10	10 (100%)	10 (100%)	10.4	319
The Netherlands	20	19 (95%)	12 (60%)	2	308
Portugal	46	40 (87%)	1 (10%)	1.05	290
Spain	33	28 (85%)	10 (56%)	1.71	239
Belgium	13	8 (62%)	6 (55%)	3.63	162
United Kingdom	50	21 (42%)	4 (9%)	1.24	103
Italy	8	8 (100%)	4 (57%)	1.88	101
Croatia	5	5 (100%)	3 (60%)	1.6	86
Germany	4	4 (100%)	3 (75%)	3	80
Switzerland	1	1 (100%)	1 (100%)	5	30
Poland	2	2 (100%)	0 (0%)	1	10
France	2	1 (50%)	0 (0%)	1	6
Serbia	1	1 (100%)	0 (0%)	1	1
Total	195	148 (76%)	54 (39%)	2.23	1735

**Table 2 insects-12-00813-t002:** Statistics of model parameters for the model explaining the number of observations during the transect walk.

Model Parameter	Edf ^1^	Chi^2^ ^2^	p ^3^
s(moy): North	2.856	56.928	<2 × 10^−16^ ***
s(moy): Southeast	2.610	32.340	2.77 × 10^−7^ ***
s(moy): Southwest	2.471	28.013	3.41 × 10^−6^ ***
s(temperature): North	1.777	40.563	<2 × 10^−16^ ***
s(temperature): Southeast	2.214	11.210	0.007 **
s(temperature): Southwest	1.848	5.470	0.081
s(start): North	1.004	2.472	0.117
s(start): Southeast	1.991	3.096	0.211
s(start): Southwest	1.000	0.043	0.835
Random effect: s(Latitude, Longitude)	2.962	82.735	<2 × 10^−16^ ***

^1^ edf: estimated degrees of freedom, ^2^ Chi^2^: chi-squared test results, ^3^ p: *p*-value with; **: <0.01 and ***: <0.001.

## Data Availability

Data is available as Appendix A.

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
