# Peer review of "The European Stag Beetle (Lucanus cervus) Monitoring Network: International Citizen Science Cooperation Reveals Regional Differences in Phenology and Temperature Response"

_insects, 2021, doi:10.3390/insects12090813_

Round 1
Reviewer 1 Report
Dear Sir, I read the manuscript titled: The European Stag Beetle (Lucanus cervus) Monitoring Net-work: international citizen science cooperation reveals regional differences in phenology and temperature response. Authors: Arno et al.
In this paper the authors report on the importance of international cooperation in monitoring of threatened species showing initial findings of the ESBMN used for Lucanus cervus monitoring. The Citizen science approach is very important in this context and in the monitoring critical species and in this direction the authors discuss on the operative protocol used in many countries that, as reported here, needs to be improved and standardized. I think that the manuscript represents a good contribute and the suggestions provided and solutions listed by the authors they can help the participants to the project to improve their investigations for a better overall result.
I absolutely agree that international standardization of the monitoring protocol and cooperation can strongly help to gather comparable data at range level of the species and the authors have clearly highlighted this focus in in this paper.
Some suggestions are listed below:
Summary
Line 40: please replace “crucial” with “vital”
Line 52: as above.
Results
Table 2: please add in legend the mening of Chi2 . Not all readers are familiar with statistical symbology.
Appendix A: Protocol
Line 494: Please pay attention to the page that you suggest. the link leads to a Chinese-language game site.
Author Response
Reviewer 1
Dear Sir, I read the manuscript titled: The European Stag Beetle (Lucanus cervus) Monitoring Net-work: international citizen science cooperation reveals regional differences in phenology and temperature response. Authors: Arno et al.
In this paper the authors report on the importance of international cooperation in monitoring of threatened species showing initial findings of the ESBMN used for Lucanus cervus monitoring. The Citizen science approach is very important in this context and in the monitoring critical species and in this direction the authors discuss on the operative protocol used in many countries that, as reported here, needs to be improved and standardized. I think that the manuscript represents a good contribute and the suggestions provided and solutions listed by the authors they can help the participants to the project to improve their investigations for a better overall result.
I absolutely agree that international standardization of the monitoring protocol and cooperation can strongly help to gather comparable data at range level of the species and the authors have clearly highlighted this focus in in this paper.
Some suggestions are listed below:
Thanks for the positive feedback.
Summary
Line 40: please replace “crucial” with “vital” accepted
Line 52: as above. accepted
Results
Table 2: please add in legend the mening of Chi2 . Not all readers are familiar with statistical symbology. Added
Appendix A: Protocol
Line 494: Please pay attention to the page that you suggest. the link leads to a Chinese-language game site. Updated, this was indeed an old link that we used but seems to be no longer valid.
Reviewer 2 Report
Dear Authors,
Thank you! Good paper!
Before publishing small revisions.
Some comments & remarks:
-Line 79: IUCN, please could you write it open also.
-Line 135: 14 countries. Pleae could you mention these countries here.
-Line 182: Italian protocol. Do we need some reference here?
-Did you collect some data also on the volunteers? For instance, the gender and age of volunteer?
-Discussuion chapter. I suggest to add some sub-headings to your Discussion:
4.1. The resusts of the analysis
4.2. The major implications of our experiences
4.3. The challenges and benifits of our approach
Author Response
Reviewer 2
Dear Authors,
Thank you! Good paper! Thanks for the positive feedback.
Before publishing small revisions.
Some comments & remarks:
-Line 79: IUCN, please could you write it open also. changed
-Line 135: 14 countries. Pleae could you mention these countries here. I have added ‘(see results)’ as they are listed there in a table and on a map.
-Line 182: Italian protocol. Do we need some reference here? Reference (Bardiani et al. 2017) included
-Did you collect some data also on the volunteers? For instance, the gender and age of volunteer? Till now this kind of data was not systematically collected but on the improved website, volunteers can now give more details concerning their profile. No changes made to the ms.
-Discussuion chapter. I suggest to add some sub-headings to your Discussion:
4.1. The resusts of the analysis
4.2. The major implications of our experiences
4.3. The challenges and benifits of our approach
Accepted
Reviewer 3 Report
General comments
The manuscript deals with an initial report about the first findings of the European Stag Beetle Monitoring Network (ESBMN) and an analysis of these results for organization and managing of citizen science cooperation. This is an interesting and important topic which fits very well to the section “Insect and human societies” of the journal “Insects”. I am not a native speaker, but I have the impression that the manuscript is written in good English. However, I recommend a check regarding consistency in using American English or British English. The title well reflects its contents. The abstract of the manuscript is informative. Research aims and hypotheses are well formulated. Material and methods are generally sufficiently described. However, I see some major flaws in the description of the research results (see specific comments).
The list of references includes 53 positions, which seem to cover well the knowledge about the research topic. The paper is supplemented by 2 tables, 5 figures and 2 appendices, all of them important for the manuscript.
Specific comments
1) Materials and Methods, Statistics, lines 185-186: The first sentence of the description of statistical methods is a bit confusing. What is the difference between “analysis” and “further analysis”. It would be helpful for the reader if the authors will specify this difference more clearly.
2) Materials and Methods, Statistics: Which software was used for realization of the statistical analyses? Please specify.
3) Results, lines 251-252: The authors state that “the number of observations was significantly influenced by all model parameters except for the starting time in the southwest region”. However, in my opinion this is an incorrect statement, because table 2 shows non-significant results for the starting time for all regions and also a non-significant result for the temperature in the southwest region. Please compare the statement cited above also with line 270 and line 279, which both corroborate table 2.
4) Results: The authors included the position of the transect (latitude and longitude) as spatial Gaussian random effect. A significant result for it is shown in table 2. However, there are no words about it in the text of the results chapter (and also not in the discussion). The reader of the paper may ask why this result is presented and what is its significance for the presented research. Therefore, I propose to add at least a short paragraph at the end of the results chapter and to discuss the significance of this result.
5) Discussion, lines 388-389: The authors write “volunteers will mainly walk transect near their home and consequently monitor urban populations”. This sentence implies that the volunteers are predominantly townspeople and not people living in rural areas. Is this really true? In lines 410-411 the authors only write that there might be an overrepresentation of urban transects in the ESBMN. However, it would be interesting to know if the composition of volunteers indeed confirms this assumption.
Conclusion
The authors present a paper, which is interesting and scientifically important. However, I think some improvements regarding the results description are necessary. I encourage the authors to consider also my comments regarding other parts of the manuscript.
Author Response
Reviewer 3
General comments
The manuscript deals with an initial report about the first findings of the European Stag Beetle Monitoring Network (ESBMN) and an analysis of these results for organization and managing of citizen science cooperation. This is an interesting and important topic which fits very well to the section “Insect and human societies” of the journal “Insects”. I am not a native speaker, but I have the impression that the manuscript is written in good English. However, I recommend a check regarding consistency in using American English or British English. Some issues with American English have been improved. The original paper was already checked by three native speaking co-authors and by spell check of MS Word. This was done again very thoroughly and American English was changed to British English and furthermore some textual improvements were made.
The title well reflects its contents. The abstract of the manuscript is informative. Research aims and hypotheses are well formulated. Material and methods are generally sufficiently described. However, I see some major flaws in the description of the research results (see specific comments).
The list of references includes 53 positions, which seem to cover well the knowledge about the research topic. The paper is supplemented by 2 tables, 5 figures and 2 appendices, all of them important for the manuscript.
Thanks for the positive feedback. We have improved the description of the results concerning your suggestions.
Specific comments
1) Materials and Methods, Statistics, lines 185-186: The first sentence of the description of statistical methods is a bit confusing. What is the difference between “analysis” and “further analysis”. It would be helpful for the reader if the authors will specify this difference more clearly. This sentence was indeed unclear as there is indeed only one analysis done on the data. The sentence now reeds: ‘Analysis was based on all transects but only transect walks ‘according to protocol’ were included (Appendix B for the data).’
2) Materials and Methods, Statistics: Which software was used for realization of the statistical analyses? Please specify. Accepted, the following sentence was added: We fit the model in R 4.0.2 [34] using the mgcv package [35].
3) Results, lines 251-252: The authors state that “the number of observations was significantly influenced by all model parameters except for the starting time in the southwest region”. However, in my opinion this is an incorrect statement, because table 2 shows non-significant results for the starting time for all regions and also a non-significant result for the temperature in the southwest region. Please compare the statement cited above also with line 270 and line 279, which both corroborate table 2. Indeed, this sentence needed to be updated and now reads: ‘The number of observations was significantly influenced by most model parameters except for the starting time and temperature effect in the Southwest region (Table 2).’
4) Results: The authors included the position of the transect (latitude and longitude) as spatial Gaussian random effect. A significant result for it is shown in table 2. However, there are no words about it in the text of the results chapter (and also not in the discussion). The reader of the paper may ask why this result is presented and what is its significance for the presented research. Therefore, I propose to add at least a short paragraph at the end of the results chapter and to discuss the significance of this result. A very good suggestion. The following text has been added: Besides the effect of the three regions, an important effect of longitude and latitude remained in the model as random effect. This random effect takes up spatial variability between transects that is not explained by other model parameters such as local habitat suitability or climate.
5) Discussion, lines 388-389: The authors write “volunteers will mainly walk transect near their home and consequently monitor urban populations”. This sentence implies that the volunteers are predominantly townspeople and not people living in rural areas. Is this really true? In lines 410-411 the authors only write that there might be an overrepresentation of urban transects in the ESBMN. However, it would be interesting to know if the composition of volunteers indeed confirms this assumption. Indeed a good point. The situation strongly depends on country and region. We assume that in some regions this bias occurs but not as strong as was interpreted here based on the original ms. Unfortunately, it hasn’t been analysed and it should be compared to the local distribution of the species as in some regions/countries most stag beetle populations are in urban while in other regions and countries it is dominantly found in natural areas. The text was clarified: The type of transects that professionals and volunteers cover might also differ. Volunteers will often walk transects near their home and consequently are well placed to monitor urban or residential populations [10]. While transects walked by professionals are more often characterised by large populations in natural and protected areas. Consequently, whether the best strategy is to rely on either volunteers or professionals or a combination of both transect types might differ depending on the local habitat. This is especially relevant to give a representative trend as it might differ between urban and protected areas.
Conclusion
The authors present a paper, which is interesting and scientifically important. However, I think some improvements regarding the results description are necessary. I encourage the authors to consider also my comments regarding other parts of the manuscript.
Thanks for the helpful review.